# Ordered Clusters of the Complete Oxidative Phosphorylation System in Cardiac Mitochondria

**DOI:** 10.3390/ijms22031462

**Published:** 2021-02-02

**Authors:** Semen Nesterov, Yury Chesnokov, Roman Kamyshinsky, Alisa Panteleeva, Konstantin Lyamzaev, Raif Vasilov, Lev Yaguzhinsky

**Affiliations:** 1Kurchatov Complex of NBICS-Technologies, National Research Center Kurchatov Institute, 123182 Moscow, Russia; chessyura@yandex.ru (Y.C.); kamyshinsky.roman@gmail.com (R.K.); raifvasilov@mail.ru (R.V.); 2Moscow Institute of Physics and Technology, 141701 Dolgoprudny, Russia; yag@genebee.msu.ru; 3Institute of Cytochemistry and Molecular Pharmacology, 115404 Moscow, Russia; 4Shubnikov Institute of Crystallography, Federal Scientific Research Centre “Crystallography and Photonics” of Russian Academy of Sciences, 119333 Moscow, Russia; 5Belozersky Research Institute for Physico-Chemical Biology, Lomonosov Moscow State University, 119992 Moscow, Russia; jalice@yandex.ru (A.P.); lyamzaev@gmail.com (K.L.)

**Keywords:** mitochondria, supercomplex, respirasome, ATP synthase, oxidative phosphorylation, cryo-electron microscopy

## Abstract

The existence of a complete oxidative phosphorylation system (OXPHOS) supercomplex including both electron transport system and ATP synthases has long been assumed based on functional evidence. However, no structural confirmation of the docking between ATP synthase and proton pumps has been obtained. In this study, cryo-electron tomography was used to reveal the supramolecular architecture of the rat heart mitochondria cristae during ATP synthesis. Respirasome and ATP synthase structure in situ were determined using subtomogram averaging. The obtained reconstructions of the inner mitochondrial membrane demonstrated that rows of respiratory chain supercomplexes can dock with rows of ATP synthases forming oligomeric ordered clusters. These ordered clusters indicate a new type of OXPHOS structural organization. It should ensure the quickness, efficiency, and damage resistance of OXPHOS, providing a direct proton transfer from pumps to ATP synthase along the lateral pH gradient without energy dissipation.

## 1. Introduction

Respiratory supercomplexes (respirasomes) in mitochondria are the keystones ensuring efficient electron transport and the absence of energy leakage. At the same time, the energy transfer to the final stage of the oxidative phosphorylation system (OXPHOS) requires protons rather than electrons. However, while the coupling between proton pumps and ATP synthase has been proposed based on functional analysis [1] and in vivo pH measurements [2], the existence of complete OXPHOS supercomplexes has not been confirmed. Here, we aim to provide the evidence for the structural connection between ATPases and respirasomes in mitochondria.

After transmembrane transfer, protons do not immediately detach from the membrane [3] due to the existence of a kinetic barrier [4,5], therefore OXPHOS clustering can be energetically beneficial. Protons on the membrane surface have a high lateral mobility [6,7,8,9], which enables their transfer for the short distances along the membrane without detachment into the bulk phase. The induction of ATP synthesis by the excess protons on the interface boundary was shown for the first time in the octane-water model system [10]. Later, the participation of laterally transferred protons on ATP synthesis was reported in mitochondria and mitoplasts [11,12]. Keeping protons on the membrane surface allows mitochondria to synthetize ATP under lower pH gradient [2]. Recently, it was confirmed that the distance between the proton pump and ATP synthase affects the speed and efficiency of ATP synthesis [13]. These data support the idea that to reach the maximal efficiency, all parts of the OXPHOS should be clustered.

The existence of respiratory supercomplexes was shown for the first time in bacteria [14] and were later detected in different species by inhibitory analysis [15] and by blue-native gel electrophoresis with mild detergents [16,17]. The last method has also shown that the long-established functional link between succinate dehydrogenase and ATP synthase [1] is accompanied by a structural link [18]. It has also confirmed the docking of some mitochondrial dehydrogenases with complex I [19], which was previously demonstrated in the isolated enzymes [20,21]. By now, it has become generally accepted that at least part of the OXPHOS can be organized as a supercomplex [22,23]. All previously described OXPHOS supercomplexes can be divided into two groups. The first is called respirasome and includes respiratory chain complexes in different proportions [24,25,26,27]. The second is called ATP-synthasome and includes ATP synthase, phosphate, and ADP transporters [28,29,30]. The direct structural coupling of these two subsystems (producer and consumer of excess protons) has not been revealed yet.

To solve this long-standing problem, we used cryo-electron tomography (cryo-ET). The resolution of single particle cryo-ET (SPT) is limited by the thickness of the sample, therefore, we studied the samples obtained through self-destruction of the rat heart mitochondria isolated by the standard protocol. The mitochondria were not exposed to any influences that could have seriously violated their native lipid and protein organization such as severe osmotic shock, detergents, or centrifugation prior to flash-freezing. The implemented approach allows for the determination of different mitochondrial complexes localized in the minimally damaged inner mitochondrial membranes as well as to obtain their structures with a satisfactory resolution.

## 2. Results

### 2.1. Respirasome Structure

After the vitrification procedure, the thinnest areas of the prepared electron microscopy grids with fragments of phosphorylating mitochondria, consisting mostly of destroyed mitoplasts and tubular cristae forming a continuous network (Appendix A), were selected for SPT. In the first step of the research, the protein composition of respirasomes and mutual arrangement of its components (complexes I, III, IV) were studied (Figure 1). Density maps with a 27–30 Å spatial resolution were obtained and two types of respirasomes with different number of complexes IV (Appendix A) were revealed. Comparison of the obtained results with blue-native electrophoresis (Appendix A) and the pertinent literature [24] indicates that the structure of the observed supercomplexes is well preserved. This was confirmed by a much higher relative content of respirasomes containing two complexes IV in our cryo-ET experiments (more than 50%).

### 2.2. The Structure of Oligomeric Oxidative Phosphorylation System (OXPHOS) Clusters

The second step included studying the supramolecular organization of mitochondrial cristae. It has been shown that under experimental conditions, the respirasomes and ATP synthases form highly ordered oligomeric elongated structures (Figure 2A). The cryo-ET data demonstrated that linear structures consisting of respirasomes I_1_III_2_IV_2_ and I_1_III_2_IV_1_ were colocalized with parallel linear structures consisting of ATP synthase oligomers (Figure 2, Figure 3 and Appendix A). The distance between respirasomes and ATP synthases in most of the observed structures was 1–5 nm. Moreover, several areas demonstrated tight docking of ATP synthases with complexes I or IV (comparable with the distance between complexes in respirasomes).

In the condensed OXPHOS clusters, the elongated transmembrane parts of complexes I are tended to orient approximately along the row of ATP synthases. While their relative position is undoubtedly not strictly fixed, statistics show that complexes I have forbidden orientations–their elongated membrane parts are virtually never oriented perpendicular to a row of ATPases (Appendix A). The cryo-ET sections and the 3D reconstructions of cristae are shown in the video (Video S1). It should be noted that in all reconstructions, only the reliably identified respirasomes and ATP synthases were demonstrated.

## 3. Discussion

Cumulatively, our data showed a new example of a possible macromolecular architecture of OXPHOS. We detected the coupling of proton pumps and ATP synthases, which occurs not at the level of individual enzymes, but via attraction of large ordered oligomeric structures that results in the formation of condensed protein clusters. The existence of these tightly docked structures still raises the question of their functional significance and operation mechanisms to be discussed further. The distinctive peculiarity of the observed structure is that it perfectly fits for a direct proton transfer from proton pumps to the ATP synthase.

The existence of stable ATP synthase rows in mitochondrial membranes was previously shown by various methods [31,32,33,34]. The possibility of linear structure formation from respirasomes was assumed earlier on the basis of structural analysis [35], but it was not proved experimentally. Moreover, there is no experimental evidence of a structural connection between respirasomes and ATP synthases.

In the discovered oligomeric structures, respirasomes are tightly docked to ATP synthases, which indicates OXPHOS clustering. It remains unclear as to what extent the discovered ordered orientation of respirasomes is important for the OXPHOS functioning in the heart and whether it manifests itself in mitochondria of other tissues. It should also be noted that this study only showed the existence of ordered OXPHOS clusters. The assessment of their abundance in intact mitochondria or in vivo requires additional research.

Due to the short distance between ATPases and proton pumps, it is tempting to identify these structures as transiently formed complete OXPHOS supercomplexes. It should be emphasized that the contact between enzymes in clusters is labile. This substantially distinguishes the observed structures from classical supercomplexes with fixed protein contacts. The linear structure is well suited for a lateral H^+^ activity (pH) gradient formation, which directs protons from respirasomes to ATP synthases. It can be assumed that clustering occurs by a mechanism similar to the formation of membrane rafts due to the special curvature and lipid composition of the membrane in the vicinity of ATP synthases. It is very likely that cristae-organizing protein opa1 [36] is also necessary for OXPHOS clustering. It can be hypothesized that clustering temporarily occurs during dynamic OXPHOS functioning. In this case, the effectiveness of OXPHOS may be controlled by the transition between a highly ordered oligomeric state with minimal proton leaks and a less ordered diffusely distributed state. Further research is needed to unveil the mechanisms of this regulation and its physiological significance. Characterization of the regulatory system that provides OXPHOS clustering under high energy demand conditions will have important applications in medicine and provide a better understanding of the pathologies associated with impaired mitochondrial function including chronic inflammatory processes, age-related diseases, ischemia, and heart failure. The development of this line of research may lead to the creation of a new class of mitochondria-targeted drugs stabilizing OXPHOS macromolecular architecture.

## 4. Methods

### 4.1. Mitochondria Isolation

Adult Wistar rats weighting 180–200 g (9–10 weeks old) were used to isolate mitochondria. Rats were housed in a conventional (non-sterile) vivarium at Moscow State University with free access to drinking water. All experiments using laboratory animals were carried out in accordance with the recommendations of the local ethics committee (in accordance with directive 2010/63/EU). All reagents used had a high purity and were made by reliable manufacturers (Merck, Kenilworth, NJ, USA; MilliporeSigma, St. Louis, MO, USA; Amresco, Solon, OH, USA; SERVA Electrophoresis GmbH, Heidelberg, Germany). Cardiac mitochondria were chosen for this study because heart tissue always experiences the energy load, so cardiac OXPHOS is more likely arranged as supercomplexes. Rats were sacrificed by decapitation without the use of anesthetics because they can potentially affect the structure of biomembranes. The time from decapitation to heart cooling was about 30 s. The isolation of subsarcolemal cardiac mitochondria was carried out according to a standard method with minor modifications. Briefly, the heart ventricles of two rats were sheared with scissors, washed from the blood, and homogenized by a Teflon pestle homogenizer in an isolation medium (220 mM mannitol, 50 mM sucrose, 5 mM EGTA, 0.1% BSA, 30 mM HEPES/KOH, pH 7.4). The resulting homogenate was centrifuged at 4 °C at 700× *g* for 8 min, then the supernatant was collected and centrifuged at 9000× *g* for 10 min. The mitochondrial fraction obtained in the sediment was resuspended in the washing medium (as an isolation medium, but without BSA and EGTA) and centrifuged at 5000× *g*. The resulting suspension of heavy mitochondrial fraction was used for cryo-electron microscopy sample preparation and blue-native gel electrophoresis. 

### 4.2. Blue Native Gel Electrophoresis 

The composition of supercomplexes after the addition of a mild detergent was analyzed by blue native polyacrylamide gel electrophoresis (BN-PAGE), as described by Schägger and von Jagow [37] but with minor modifications. Mitochondrial proteins from the heart tissue were solubilized with recently recrystallized digitonin (1 g/g), then Coomassie blue G-250 (2.5 µL of a 5% stock in 500 mM 6-aminocaproic acid) was added to the samples. BN-PAGE was carried out in gradient 4–15% Bis-Tris gels. Both stacking and resolving gels contained 0.025% digitonin. The gels were run at 4–7 °C in the Bio-Rad Mini-PROTEAN Tetra system at 100–250 V within 3–6 h. We used the anode buffer (50 mM Bis-Tris with pH 7.0) and the cathode buffer (50 mM Tricine, 15 mM Bis-Tris, and 0.002% Coomassie blue G-250 with pH 7.0). As opposed to the original method for BN-PAGE, we did not change the cathode buffer during electrophoresis and it contained a lower concentration of Coomassie blue. After electrophoresis, the gels were stained with Coomassie blue G-250 (0.05% Coomassie, 40% ethanol, 10% acetic acid), which was followed by a destaining step. We determined the size of the complexes using the SERVA Native Marker. The presence of complex I in all bands that corresponded to supercomplexes was confirmed by the in-gel complex I activity assay [38].

### 4.3. Sample Preparation for Cryo-Electron Microscopy

The dense mitochondrial suspension was diluted to a concentration of ~0.3 mg/mL in a respiration medium (KCl 80 mM, MgCl2 1 mM, HEPES/KOH 20 mM, KH2PO4 10 mM, pH 7.4) and stored for 1 hour without exogenous substrates in a closed microtube on ice (about 4 °C). Then, the mitochondria were slowly heated to 20 °C. Ten minutes prior to vitrification, phosphorylation was started by adding 10 mM glutamate, 4 mM malate, and 2 mM ADP. Then, 20 μL of the obtained sample was mixed with 1 μL gold nanoparticle solution (10 nm Colloidal Gold Labeled Protein A, UMC Utrecht, The Netherlands) and 3 μL of the mixture was applied to a glow-discharged (30 s, 25 mA) Lacey Carbon EM grid. After blotting for 2.5 s at 4 °C, the grid with the specimen was plunge-frozen into liquid ethane in Vitrobot Mark IV (Thermo Fisher Scientific, Hillsboro, OR, USA).

### 4.4. Cryo-Electron Tomography

The study was carried out with a Titan Krios 60–300 TEM/STEM (Thermo Fisher Scientific, Hillsboro, OR, USA) cryo-electron microscope, equipped with a Falcon II direct electron detector (Thermo Fisher Scientific, Hillsboro, OR, USA) and Cs image corrector (CEOS, Germany) at an accelerating voltage of 300 kV. Eleven tilt-series of the sample were collected automatically with Tomography software (Thermo Fisher Scientific, Hillsboro, OR, USA) in low-dose mode with 18,000× magnification (pixel size 3.7 Å) and the defocus value in the range between −6 and −8 μm using the bidirectional tilt scheme (0°, −2°, …, −58°, −60°, 2°, 4°, …, 58°, 60°). The accumulated total dose was ~100 ē/Å^2^.

### 4.5. Tomogram Reconstruction

Cross-correlation alignment and tomographic reconstruction were performed using the IMOD (version 4.9.10, University of Colorado, Boulder, CO, USA) software package [39] by the simultaneous iterative reconstruction technique (SIRT) and weighted back-projection (WBP) method [40]. Gold nanoparticles were used as fiducial markers for the alignment of the tilt-series projection images. Two tilt series with the alignment error not exceeding 1 pixel were selected for further processing. The reconstructions obtained by SIRT were used for visual inspection and particle picking, while the WBP reconstructions were used for sub-tomogram averaging.

### 4.6. Segmentation and Sub-Tomogram Averaging

The coordinates of 100 supercomplexes (“particles”) with clearly visible I_1_ and III_2_ were manually picked from tomographic sections with IMOD and utilized for de novo reconstruction in Relion2 (University of Cambridge, UK) [41] using sub-tomogram averaging (protocol described in [42]), which resulted in a 44 Å resolution, as determined by the Fourier Shell Correlation (FSC) 0.143 criteria. The same particles were used as positive examples for automated segmentation using convolutional neural network utility [43] in an open-source EMAN2.23 package (Baylor College of Medicine, Houston, TX, USA) [44] on two times binned data (pixel size 7.4 Å) restored with SIRT. ATP synthases, pyruvate dehydrogenase complexes, gold fiducials, edges of the carbon support, and contamination on the grid surface were used as negative examples for neural network training.

To find the coordinates of supercomplexes on segmented tomograms, the “Find particles from segmentation” utility in EMAN2.23 was used. A total of 5821 automatically picked sub-tomograms were extracted from the two times binned dataset (pixel size 7.4 Å) restored with WBP, aligned to a de novo model using 3D auto-refinement (Relion2) and sorted into 3D classes without alignment (τ2_fudge = 8). One class with 1706 particles was chosen for further processing. Refinement with low-pass filtered to a 60 Å de novo model as the reference resulted in a 24.6 Å resolution (FSC = 0.143) supercomplex density map. 

Since the densities of the IV complexes varied, an additional focused 3D classification (τ2_fudge = 20) was performed using a mask on complex IV. A total of 944 particles (55%) consisting of I_1_III_2_IV_2_ resulted in a 27.7 Å density map (EMD-11605), and 762 particles (45%) corresponding to I_1_III_2_IV_1_, resulted in a 29.6 Å density map (EMD-11604). A 14.8 Å density map of ATP synthase dimers was obtained from the same dataset as described previously [31]. 

Graphics, final visualization, and fitting were performed with UCSF Chimera [45]. The PDB-5xtd, the PDB-1bgy, and the PDB-5z62 structures from the Protein Data Bank were used for the fitting of the I, III, and IV complexes, respectively. Mitochondrial membranes were manually segmented with the Avizo (Thermo Fisher Scientific, Hillsboro, OR, USA) software package.

## 5. Conclusions

Here, a new type of the OXPHOS structural organization was revealed as a condensed cluster formed of roughly parallel rows of respirasomes and ATP synthases. The short distance between proton pumps and ATP synthases theoretically allows for the direct transfer of hydrogen ions to ATP synthases with maximum speed and minimal leakages [13]. We believe that observed linear OXPHOS clusters are optimal structures for providing kinetic coupling [2] between respirasomes and ATP synthases through lateral H+ activity (pH) gradient formation. Thus, the discovered structures may provide both quickness and efficiency of ATP synthesis, which are essential for heart mitochondria under high load conditions. Dynamic formation and dissipation of such clusters can be one of the mechanisms for synchronization of the OXPHOS components and their activity regulation.

## Figures and Tables

**Figure 1 ijms-22-01462-f001:**
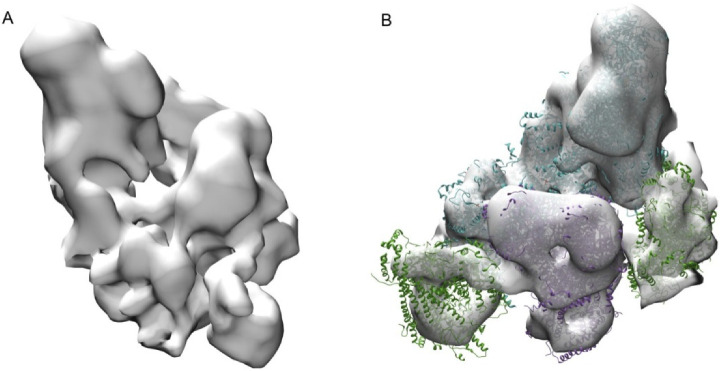
(**A**) Cryo-electron microscopy map of the most common type of respirasome (mitochondrial respiratory chain supercomplex). (**B**) 90° rotated cryo-EM map with fitted high-resolution structures (PDB 5z62, 5xtd, 1bjy). Blue–complex I, purple–complex III dimer, green–complex IV.

**Figure 2 ijms-22-01462-f002:**
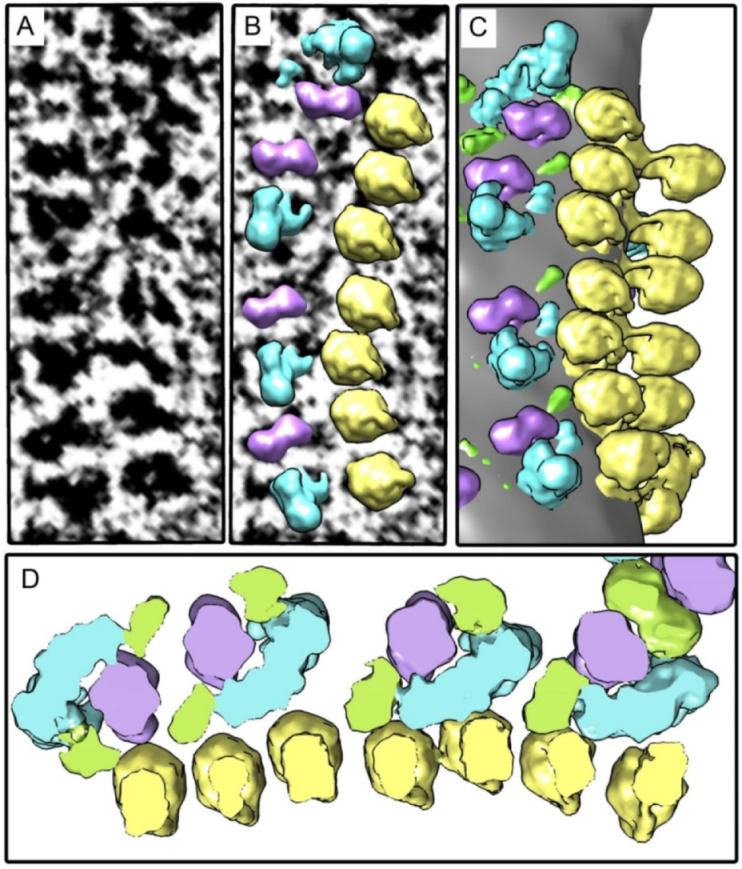
An example of the oligomeric linear structure consisting of tightly docked ATP synthases and respirasomes. Elongated transmembrane parts of complexes I are oriented approximately parallel to the row of ATP synthases. (**A**) Tomographic slice; (**B**) Density maps of complexes I, III_2_, and ATP synthases placed back to the tomogram; (**C**) Surface rendering of the selected area; (**D**) Slice of the density map, view from the intermembrane space. Image (**D**) is a bit enlarged in comparison with images (**A**–**C**). Colors: yellow–ATP synthase, blue–complex I, purple–complex III dimer, green–complex IV.

**Figure 3 ijms-22-01462-f003:**
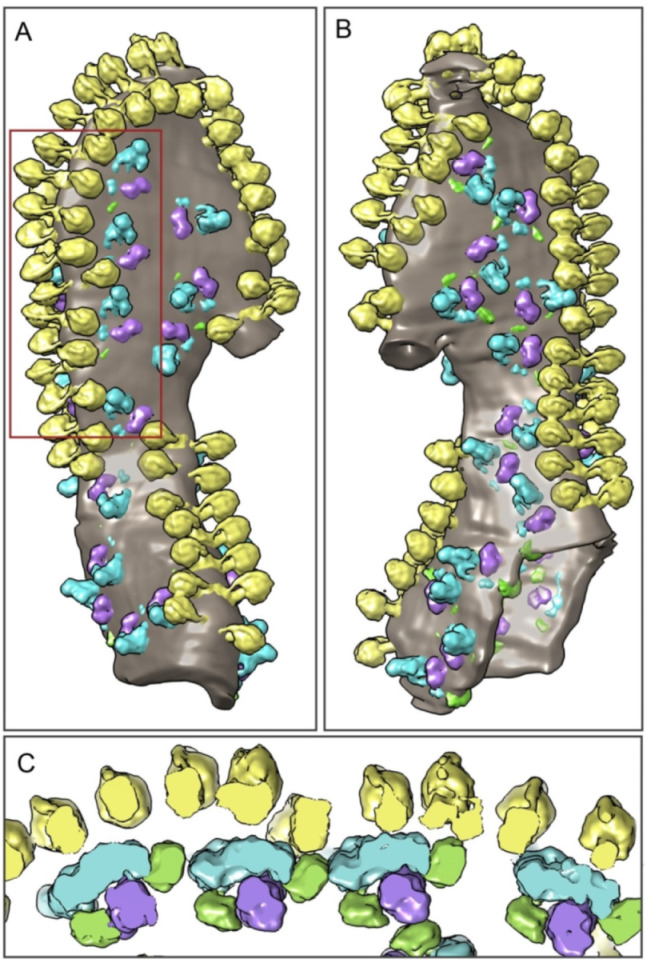
Surface rendering of the mitochondrial crista containing the oligomeric linear structure consisting of the respiratory chain and ATP synthases. (**A**,**B**) Views of the crista from the opposite sides (180° rotation around the vertical axis), demonstrating a highly ordered structure of the hypercomplex; (**C**) 3D visualization of respirasomes and the ATP synthases contact zones in the membrane. Slice of the fragment highlighted by the red frame in image (**A**). View from the opposite side of the membrane. Colors: yellow–ATP synthase, blue–complex I, purple–complex III dimer, green–complex IV.

## Data Availability

All data are available in the manuscript or the Appendix A, electron microscopy density maps are deposited in the EMDB (EMD-11605, EMD-11604).

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
