# Peer review of "Ordered Clusters of the Complete Oxidative Phosphorylation System in Cardiac Mitochondria"

_ijms, 2021, doi:10.3390/ijms22031462_

Round 1

Reviewer 1 Report

In this study the authors have used cryo-electron tomography to examine the structure and location of OXPHOS complexes and ATP synthase, particularly in relation to each other.  This study adds to previous knowledge obtained with other techniques.

The cryo-EM method appears to be well described, however, the manuscript would benefit from the following information being added to the methods:

  • How were the rats housed and killed?
  • Why were cardiac mitochondria examined?
  • How many rats were used and how many replicates were examined i.e. biological and technical replicates
  • Please go into more detail regarding statistical analysis, particularly for figure S5.

Please add arrows or some other type of markers to figure S1 to show the tubular cristae for clarity.

Figure S5 is not very clear, should there be error bars?  Which bars correspond with which inserts along the top?

Reviewer 2 Report

This study provides new information the complete oxidative phosphorylation system in cardiac mitochondria. A major revision is suggested.

  1. The abstract needs a massive revision and improve. Please address, background, methods and conclusion of this study.
  2. Please address clinical implications of this study.
  3. Please provide details of animal study. How to scarify? Anesthetics used?
  4. How many animals were used? How many repetition of each investigation? Please address.
  5. Please discuss limitations of this study.

Round 2

Reviewer 2 Report

Paper is fine. Authors made all necessary corrections. Thanks